# IoT Wheelchair Control System Based on Multi-Mode Sensing and Human-Machine Interaction

**DOI:** 10.3390/mi13071108

**Published:** 2022-07-15

**Authors:** Jianwei Cui, Linwei Cui, Zizheng Huang, Xiang Li, Fei Han

**Affiliations:** Institute of Instrument Science and Engineering, Southeast University, Nanjing 210096, China; 220203464@seu.edu.cn (L.C.); 220213616@seu.edu.cn (Z.H.); 220213755@seu.edu.cn (X.L.); 220213750@seu.edu.cn (F.H.)

**Keywords:** intelligent wheelchair, multimode sensing, Internet of things, human-machine interaction, remote control

## Abstract

Traditional wheelchairs are unable to actively sense the external environment during use and have a single control method. Therefore, this paper develops an intelligent IoT wheelchair with the three functions, as follows. (1) Occupant-wheelchair-environment multimode sensing: the PAJ7620 sensor is used to recognize gesture information, while GPS (Global Positioning System) and IMU (Inertial Measurement Unit) sensors are used to sense positioning, speed and postural information. In addition, Lidar, DHT11, and BH1750 sensors obtain environmental information such as road information, temperature and humidity and light intensity. (2) Fusion control scheme: a mobile control scheme based on rocker and gesture recognition, as well as a backrest and footrest lifting, lowering and movement control scheme based on Tencent Cloud and mobile APP (Application). (3) Human-machine interaction: the wheelchair is docked to Tencent IoT Explorer through ESP8266 WiFi module, using MQTT (Message Queuing Telemetry Transport) protocol is used to upload sensory data, while the wheelchair status can be viewed and controlled on the APP. The wheelchair designed in this paper can sense and report the status of the occupant, environment and wheelchair in real time, while the user can view the sensory data on the mobile APP and control the wheelchair using the rocker, gestures and APP.

## 1. Introduction

The problem of an ageing population is becoming increasingly serious, with disabled people accounting for around 15% of the world’s population, of whom 285 million are visually impaired or blind, Wheelchairs are an indispensable travel aid for people with disability, with annual demand for wheelchairs reaching over 30 million. At the same time, with the widespread use of wheelchairs, wheelchair users and guardians have become more concerned about the comfort and safety of wheelchairs, which has led to the rapid development of research on smart wheelchairs [1]. Smart wheelchairs, in which cameras, lidar and EEG (electroencephalogram) sensors are installed on traditional wheelchairs to build sensor networks, provide users with a full range of sensing, human-machine interaction, remote monitoring and mobility control functions, are smarter and safer to use than traditional electric or push wheelchairs, and are the future direction of wheelchairs.

The perception function of the intelligent wheelchair includes the perception of the external environment and the perception of the user. In terms of wheelchair occupant perception, Zhang Zhen constructed a safety monitoring system for wheelchairs, installed sensors for positioning, pressure sensing and body temperature measurement on the wheelchair, and was able to view various types of information about the wheelchair in real time on the mobile phone [2]. Rahimunnisa studied the installation of MPU6050 and SPO2 sensors made by InvenSense, Sunnyvale, the U.S.A on the wheelchair, which were able to detect the orientation of the user’s hand and health parameters, while the data health was synchronized to the cloud for viewing [3]. Basak et al. studied a smart wheelchair based on gesture recognition by installing a gesture detection sensor on the wheelchair [4].

In the area of wheelchair environment perception, Zhanyinze et al. proposed a vehicle target recognition algorithm based on lidar and infrared image fusion, which can recognize vehicle targets [5]; Chenguang Liu et al. built an experimental platform for 3D lidar target recognition for unmanned boats in real time, solving key technologies such as laser point cloud data processing, target segmentation, and remote interaction of point cloud images [6]. Xiangmo Zhao et al. fused lidar range data and camera image data to obtain Regions of Interest using lidar, and then input to a CNN (convolutional neural network) with image data for training to achieve accurate recognition of objects around cars [7]. Radhika Ravi et al. studied a lane width recognition algorithm for moving regions, extracting road surface information from laser point clouds, extracting pavement information, extracting pavement markings and clustering them based on intensity thresholds. They used equally spaced centreline points to calculate lane widths with an accuracy of 3 cm for lane width estimation [8]. Reza et al. achieved beacon recognition on roads based on linear and kernel (non-linear) support vector machines, multi-core learning, light detection and radar ranging 3D data [9].

The movement control mode of intelligent wheelchairs includes three modes: handle control, occupant state detection and autonomous navigation movement, among which the handle control mode is for the user to control the movement direction and speed of the wheelchair through the electric handle; the occupant state detection mode is to control the movement of the wheelchair by detecting the user’s head posture, pupil centre position, electromyographic signals and EEG signals [10,11,12,13]. Wang Beiyi designed an intelligent wheelchair that combines head movement control with traditional button control; it is capable of detecting the user’s head posture to achieve wheelchair control, as well as having distance detection and alarm functions [14]. Tang Xianlun et al. installed an EEG signal acquisition module on the wheelchair, capable of collecting the user’s motion imagination EEG signals to control the wheelchair [15]. Wu Jiabao et al. installed a human visual tracking device that was able to capture and analyse the human eye position to control the wheelchair’s movement [16]. Javeria Khan investigated a wheelchair control mode based on EEG signals [17]. Rosero-Montalvo investigated a wheelchair control mode based on human posture [18]. Al-Wesabi et al. combined brain activity, blink frequency and Ardunio controller to achieve wheelchair mobility without manual control for disabled people [19]. Baiju et al. designed a smart wheelchair that included both manual and automatic control modes, using infrared sensors to assist movement in manual mode and a camera and Raspberry Pi for obstacle avoidance and image processing in automatic mode [20]. Autonomous guided mobility is achieved by installing inertial sensors, lidar, vision and range sensors on the wheelchair to sense the wheelchair’s surroundings [21,22]. Megalingam et al. installed 2D lidar on the wheelchair to enable autonomous navigation of the wheelchair within a specified area without human control [23].

This paper carries out research on the perception, control and human-machine interaction of intelligent wheelchairs, mainly accomplishing the following:

(1) Sensing occupant gesture change information through gesture recognition sensors to achieve occupant perception; detecting posture angle, positioning and speed information during wheelchair movement through posture sensors and GPS positioning sensors to achieve wheelchair state perception; using lidar and temperature and humidity sensors to identify road information such as lane lines and pedestrians and environmental conditions to achieve multi-mode information for human-wheelchair-environment sensing.

(2) Wheelchair multi-mode control solution, installing a handle on the traditional hand-push wheelchair, which can be used to control the wheelchair quickly and easily; developing a gesture control system, using different gesture changes identified by gesture sensors, combined with embedded processor and motor drive to achieve wheelchair movement control, suitable for people with hand disabilities and unable to grasp the handle; developing a remote control system for the wheelchair based on the App and the Tencent IoT Explorer. The wheelchair, Tencent Cloud and the mobile phone use the MQTT protocol to interact with sensory data and control commands, so that users and guardians can view the sensory data and control the wheelchair remotely on their mobile phones, including movement, footrest and backrest control, further improving the safety performance of the wheelchair.

The remainder of the paper is organised as follows. In Section 2, we present the general design of the intelligent wheelchair. Section 3 presents the study of multimodal sensing technology for wheelchairs, including user sensing, wheelchair state sensing and environment sensing. Section 4 presents the local and remote control of the wheelchair. Experiments with the wheelchair are presented in Section 5, and Section 6 concludes the paper.

## 2. System Overview

This paper studies the wheelchair’s multi-mode perception and control technology based on rocker, gesture recognition and mobile app, and the system is divided into five layers: remote end, main control layer, communication layer, perception layer and drive layer, as shown in Figure 1. The system is divided into a five-layer structure of remote terminal, master control layer, communication layer, perception layer and driver layer, as follows.

(1)The remote terminal includes the Tencent IoT Explorer and the mobile phone APP.(2)The master control layer adopts NXP RT1062 made by NXP, Netherlands development board and Raspberry Pi 3B+ made by Raspberry Pi, the United Kingdom development board, where the NXP RT1062 development board issues control commands and collects sensing data from the whole system It connects to Tencent IoT Explorer through ESP8266 WiFi module to upload sensing data and receive remote control commands based on MQTT protocol stack, in addition to using the network port of raspberry pie 3b+ development board to communicate with radar, obtain point cloud data, and send the results to the NXP RT1062 development board after processing.(3)The communication layer is the interface between the main control layer and the perception layer and driver layer, including WiFi, serial port, IIC and other communication interfaces.(4)The sensing layer includes three types of sensors: occupant sensing, wheelchair state sensing and environment sensing.(5)The driver layer receives commands from the main control layer and completes the movement control of the wheelchair and the raising and lowering control of the footrest and backrest.

## 3. Smart Wheelchair Multi-Mode Sensing Technology

In the process of using traditional wheelchairs, the following problems exist: (1) In terms of mobility control, the handle control is mainly for healthy people, while users with missing palms or unable to hold the handle have to rely on others to use the wheelchair. It is necessary to install an occupant detection device on the wheelchair to realise the use of the wheelchair. (2) When using a conventional wheelchair, the user’s family members do not have access to status information such as the wheelchair’s positioning in a timely manner, making it impossible to deal with emergencies experienced by the user in time. (3) The occupant needs to pay attention to the environment around the wheelchair at all times during the use of the wheelchair, which increases the burden of using the wheelchair.

In response to the problems of traditional wheelchairs, this article proposes a multimode sensing scheme for wheelchairs and a remote viewing scheme for sensing data, as shown in Figure 2. We first proposed a multimode perception technology including occupant perception, state perception and environment perception. To solve problem (1), gesture recognition sensors were installed on the wheelchair and the microcontroller received movement data from the end of the arm using the IIC protocol to achieve gesture perception; for problem (2), we installed MEMS (Micro-Electro-Mechanical System) posture sensors and positioning sensors on the wheelchair to achieve state perception; for problem (3), we installed lidar and weather sensors on the wheelchair and invoked the radar recognition algorithm to collect road information and weather data to achieve perception of the environment.

In addition, we designed a remote viewing solution for wheelchair-aware data based on the MQTT protocol. The embedded processor encapsulates the sensing data into frames and uses the WiFi module to transmit them in real time to the Tencent IoT Explorer. In addition, the mobile phone and APP is connected to the IoT Explorer and will automatically update the data on the APP interface when the data is updated. Figure 2 shows the architecture design of the multi-mode sensing solution.

### 3.1. User State Awareness Based on Gesture Sensor

People with muscle atrophy and arm weakness are unable to control the wheelchair by means of a joystick or hand push, while the joystick is flexible and unsuitable for some elderly people. Therefore, we designed a user state sensing solution based on gesture recognition, in which gesture recognition sensors are installed on the wheelchair to detect changes in the user’s gestures and use the gesture recognition information to also control the movement of the wheelchair. In addition, pulse and blood pressure sensors are also installed on the wheelchair in order to fully sense the physiological status of the user.

For recognition accuracy, cost and practicality considerations, this paper uses the high-performance gesture recognition sensor module ATK-PAJ7620 from Alientek, Guangzhou, China, which uses the PAJ7620U2 chip from Original Phase Technology (Pixart) and supports the recognition of 9 gesture types such as left, right, front, back, etc. [24]. Figure 3 shows the physical diagram of the sensor and the gestures that can be recognized.

The functional block diagram of the PAJ7602U2 chip made by PixArt, Taiwan, China is shown in Figure 4. During operation, through the internal LED (Light Emitting Diode) driver, the infrared LED is driven to emit infrared signals outwards. When the sensor array detects an object in the effective distance, the target information extraction array will acquire the characteristic raw data of the detected target, and the acquired data will be stored in the register, while the gesture recognition array will recognize the raw data for processing, and finally store the recognition result in the register, and output the recognition result using the IIC bus.

When using the PAJ7620 sensor, the IIC interface of the microcontroller is connected to the sensor and the sensor is driven in three steps: wake-up, initialisation and recognition test. The gesture information is obtained by reading and writing the two bank register areas inside the sensor during the test.

### 3.2. Wheelchair Status Awareness

#### 3.2.1. MEMS Sensor Based Wheelchair Posture Sensing

Wheelchairs can tip over during movement due to road conditions or improper handling by the user, so there is a need to sense the posture of the wheelchair and transmit posture data to the remote guardian in real time to prevent the wheelchair from tipping over without rescue. We have used the ATK-MPU6050 si*x*-axis attitude sensor made by Alientek, Guangzhou, China [25], based on the IIC (Inter-Integrated Circuit) protocol, which is capable of outputting three-axis acceleration and three-axis gyroscope information to enable real-time monitoring of the wheelchair’s attitude angle, as shown in Figure 5.

The raw data output from the MPU6050 sensor made by InvenSense, Sunnyvale, the U.S.A is acceleration data and angular velocity data, which is not intuitive for the user to view, so the article uses the Data Management Platform that comes with the MPU6050 to convert the raw data into quaternions, noted as [*q*0, *q*1, *q*2, *q*3], and then converts the quaternions into attitude angles using Equation (1).
(1)$$\begin{array}{l} pitch=asin(-2\times q1\times q3+2\times q0\times q2)\times 57.3\\ row=atan2(2\times q2\times q3+2\times q0\times q1,-2\times q1\times q1-2\times q2\times q2+1)\times 57.3\\ yaw=atan2(2\times (q1\times q2+q0\times q3,q0\times q0+q1\times q1-q2\times q2-q3\times q3)\times 57.3 \end{array}$$
where pitch is the pitch angle, yaw is the yaw angle and roll is the roll angle and *q*0, *q*1, *q*2 and *q*3 are the quaternions solved by the sensor. By using this formula, the pose angle can be solved.

#### 3.2.2. GPS Module Based Wheelchair Position Awareness

In order to help the guardian to obtain the location information of the wheelchair user in real time, the ATK-GPS/BeiDou module from Alientek, Guangzhou, China was installed [26], which uses the S1216F8 BD module made by Skytrax, London, England with a smaller size and a positioning accuracy of 2.5mCEP (SBAS: 2.0mCEP), thus helping the loved ones of wheelchair users to obtain real-time information on the location of the wheelchair.

The positioning module communicates with the external embedded controller via the serial port and outputs GPS/BeiDou positioning data including longitude, latitude and speed according to the NMEA-0183 protocol; the control protocol for the module is SkyTraq. The protocol uses ASCII codes to transfer GPS positioning information and the format of a frame of data is as follows.
$ aaccc, ddd, ddd, …, ddd * hh (CR)(LF)
where $ is the start bit, aaccc is the address bit, ddd, ..., dd is the data bit, * is the checksum prefix, hh is the checksum and (CR)(LF) is the end-of-frame flag. Figure 6 shows the ATK-S1216F8-BD positioning module and the active antenna of the SMA interface.

### 3.3. Wheelchair External Environment Awareness

#### 3.3.1. Road Surface Perception Based on 3D Lidar

In order to sense the road environment in real time and assess the safety status of the wheelchair while it is moving, the article selects RoboSense’s 3D lidar RS-LiDAR-M1 made by Robosense, Shanghai, China, which is the world’s first and only radar based on MEMS intelligent scanning technology and AI algorithm technology for self-driving passenger cars, using a main data stream output protocol to output 3D point cloud coordinate data, reflection intensity information and time stamps. Figure 7 shows a physical view of the lidar, the point cloud collected and the connection to the processor. The radar parameters are shown in Table 1 [27].

In addition, the lidar integrates road environment sensing algorithms, combining traditional point cloud algorithms based on geometric rules with data-driven deep learning algorithms. Firstly, the 3D point cloud is filtered using the downsampling method, and the large-scale point cloud is divided into a 3D grid with a number of points *k*; $\left(x_{i},y_{i},z_{i}\right)$ is the coordinate of the point and (x,y,z) is the centre-of-mass coordinate.
(2)$$\begin{array}{l} x=\sum_{i=1}^{k}x_{i}/k\\ y=\sum_{i=1}^{k}y_{i}/k\\ z=\sum_{i=1}^{k}z_{i}/k \end{array}$$

Road routes, boundaries and other objects are then extracted using geometric methods and fused with neural network algorithms to classify other objects, and to output road object sensing information in seven categories: cylinders, pedestrians, cyclists, small vehicles, large vehicles, trailers and unknown categories.

#### 3.3.2. Environment Awareness

During the movement of the wheelchair, we deployed the DHT11 temperature and humidity sensor and the BH1750 light sensor made in Asair, Guang, China to collect information on the temperature [28,29], humidity and light intensity of the environment in which the wheelchair is located, while displaying this information in the APP, so that the wheelchair user can remotely access the real-time environmental status of the wheelchair.

The sensors are shown in Figure 8, where the DHT11 sensor uses a single bus data format with 8 bit humidity integer data + 8 bit humidity fractional data + 8 bi temperature integer data + 8 bit temperature fractional data in the data packet; the BH1750 sensor uses an IIC(Inter-Integrated Circuit) interface and has a built-in 16 bit AD converter for high accuracy measurement of brightness at 1 lux.

### 3.4. Remote Viewing of Sensory Data

After the embedded processor collects multi-mode sensing data, in order to view the sensing data remotely on the mobile phone APP, a data remote viewing solution based on the Tencent IoT Explorer was designed. Tencent IoT Explorer is an IoT PaaS platform launched by Tencent Cloud for smart life and industrial applications. It supports WiFi, cellular, LoRa, Bluetooth and other communication standard devices on the cloud. Developers’ devices can be quickly uploaded to the cloud and provide efficient information processing and convenient data services. At the same time, Tencent IoT Explorer has launched Tencent Connect, which allows developers to complete application development for their devices through a development-free panel.

The data remote viewing solution includes: (1) a communication protocol, supporting two-way interaction of data between APP, embedded processor and cloud platform using MQTT server/client protocol; (2) updating and viewing of data. Firstly, the embedded processor on the wheelchair side encapsulates the collected data and uses a WiFi module to access Tencent IoT Explorer to publish sensory data packets. Meanwhile, APP subscribes to the data and accesses the cloud platform to obtain the real-time data uploaded by the wheelchair, which in turn enables viewing of data.

#### 3.4.1. MQTT-Based Protocol for Wheelchair Data Communication

The article uses the MQTT server/client protocol, with the Tencent IoT Explorer as the MQTT server and the mobile app and wheelchair as the MQTT clients. The server node is responsible for managing the data published by the client node, and the client node can also subscribe to the data on a topic held by the server.

Table 2 shows the designed communication topics, including data topic, event topic and control topic, where the content of the data topic is sensed data, the content of the event topic is events reported by wheelchairs, and the content of the control topic is control commands. When a client publishes a data topic named ‘ID/${deviceName}/data’ to the server, all clients subscribed to this topic will receive the data, thus enabling dynamic updating of the data.

#### 3.4.2. Remote Data Transmission and Encapsulation

The embedded processor needs to communicate with the MQTT server of the cloud platform to complete the reporting of sensory data, so the article installed an ESP8266 WIFI communication module at the wheelchair end, as shown in Figure 9a for the physical diagram, which is made by Espressif, Shanghai, China.

The embedded processor drives the WiFi module to access the network and accesses the HTML address of the Tencent IoT Explorer so that the wheelchair end can report data to the IoT Explorer as an MQTT client. When reporting sensory data, the embedded processor encapsulates the collected sensory data into a frame of data, as shown in Figure 9b, which is 9 bytes in total, including 3 bytes of attitude data, 3 bytes of positioning and velocity data and 3 bytes of temperature, humidity and light intensity data. Then the sensing packet with the topic ‘788H526A3U/${deviceName}/data’ is sent, where 788H526A3U is the device ID number.

Once IoT Explorer has received the packet, it sends the contents of the packet to the mobile app subscribed to this topic so that the user can view the perception data of the wheelchair on the mobile phone.

#### 3.4.3. APP Interface

The Tencent IoT Explorer integrates the Tencent Connect APP design function, which allows developers to configure the functions of the APP using a graphical interface on top of the cloud platform, thus greatly reducing the development cycle. Once the APP is created, users can scan the APP QR code and start using it. Figure 10 shows the app design interface.

Figure 11 shows the app perception information viewing interface designed in this paper, including the wheelchair’s current positioning information, speed, lighting brightness, temperature and humidity three-axis posture angle information. The interface will dynamically update the display based on the data.

## 4. Smart Wheelchair Control Technology

### 4.1. Wheelchair Mobility Control Based on Gesture Recognition

Traditional wheelchairs need to be pushed by a person to move. Electric wheelchairs add a motorised rocker, which can be turned by the occupant to move the wheelchair forward, backward and steer, while buttons next to the rocker can be used to increase or decrease the speed of the wheelchair, as shown in Figure 12. However, people with arm weakness may be unable to control the rocker, so we first designed a gesture recognition-based wheelchair control method, which controls the movement of the wheelchair through changes in the user’s arm gestures.

In Section 3.1, relying on the PAJ7620 sensor we can recognise human hand gestures and can therefore define the direction of movement of the wheelchair according to different hand gestures, specifically: (1) hand gesture forward, the wheelchair moves forward; (2) hand gesture back, the wheelchair moves backwards; (3) hand gesture left, the wheelchair turns left; (4) hand gesture right, the wheelchair turns right; (5) hand gesture down, the wheelchair stops moving.

In addition, when the embedded processor detects a change in gesture, a combination of embedded processor and motor drive module is used in order to achieve movement control of the wheelchair. One DC motor is mounted on each of the two mobile wheels of the wheelchair and each motor is connected to a motor drive module. The connection schematic is shown in Figure 13. The control interfaces of the motor drive module include IN1, IN2 and PWM, where IN1 and IN2 are used to control the steering of the motor and the PWM interface is used to control the speed of the motor. When the wheelchair moves forward, the input levels of IN1 and IN2 are 1 and 0. When the wheelchair moves backwards, IN1 and IN2 are 0 and 1. The left turn and right turn use the PWM interface to control the movement speed of the two wheels to achieve a differential turn.

### 4.2. Remote Control of Wheelchairs Based on Tencent IoT Explorer

With the development of cloud computing and IoT technology, remote control of wheelchairs has become possible. Wheelchair users can not only control their wheelchairs locally through gestures, but can also control various modules of their wheelchairs and view wheelchair sensory data at remote terminals with the help of a mobile phone APP and a cloud platform.

Figure 14 shows the intelligent wheelchair remote control scheme designed in the article. The remote control process of the wheelchair is as follows: the user clicks the control button on the mobile phone APP, the APP sends control commands to the Tencent IoT Explorer, the IoT Explorer receives the control commands and forwards them to the intelligent wheelchair connected to the cloud platform using the WIFI module, the embedded processor of the intelligent wheelchair parses the control commands and controls the motor drive module to execute backrest lift, footrest lift and wheelchair movement control.

#### 4.2.1. Design of Wheelchair Footrest and Backrest Lift Based on Electric Actuator

In order to realise the lift control of the footrest and backrest of the wheelchair, we have installed electric actuators at the footrest and backrest of the wheelchair. The electric actuators use DC brush motors, which can be controlled by supplying 0 V and 24 V voltage to the positive and negative terminals of the motor respectively to raise the actuators and vice versa to lower them.

Figure 15a shows the installation picture of the electric push rod at the foot pedal. Two foot pedals of the wheelchair are respectively installed with an electric push rod. The motor drive 1 in Figure 14 is used to drive both electric actuators, so that the footrests can be raised and lowered at the same time. Figure 15b shows the backrest with two electric actuators, which are driven simultaneously by motor drive 2 in Figure 14 to raise and lower the backrest.

#### 4.2.2. APP Control Interface

The control interface of the wheelchair was designed based on Tencent Connect in order to enable the guardian of the wheelchair to send control commands to the wheelchair remotely on the mobile phone during the use of the wheelchair, as shown in Figure 16.

Based on this control interface, after the user presses the button of the mobile app, the mobile phone as a client node will send control data with the topic ‘788H526A3U/${deviceName}/control’ to the MQTT server, where ‘788H526A3U’ is the device ID number and the control command corresponding to this topic will be stored in the MQTT server. Since the wheelchair end subscribes to this topic, the embedded processor at the wheelchair end will receive the control command on this topic in real time and execute the corresponding control action.

## 5. Experimental Result

### 5.1. Smart Wheelchair Physical Appearance

After studying various technologies such as sensing, gesture control and remote control of wheelchairs, in order to test these theories, we modified commercially available hand-push wheelchairs during our experiments and installed sensors, actuators and controllers, etc., as shown in Figure 17. The side view of the smart wheelchair is shown on the left and the front view of the smart wheelchair is shown on the right. The sensors installed on the wheelchair include gesture recognition sensors, posture sensors, GPS positioning sensors, temperature and humidity sensors and lidar. In addition, the wheelchair is fitted with an NXP RT1062 embedded processor made by NXP, Netherlands, Raspberry Pi, WiFi module, motor driver and power module. The Raspberry Pi with Ubuntu installed uses ROS to acquire laser point cloud data from the radar at a frequency of 10 Hz and sends it to the RT1062 via the serial port, while the RT1062 acquires data acquired by other sensors using timed interrupts with an acquisition interval of 100 ms.

After installing a large number of sensory sensors and processors, a 24 V 30 A lithium battery was installed in the wheelchair in order to ensure a good power supply for the system. After installation, the intelligent wheelchair weighs around 10 kg more than a conventional wheelchair and can move up to 15–20 km, which is sufficient for the user’s daily travel. We arranged for testing by 20 persons, including healthy people, disabled people and others. We first completed multimode perception experiments, including occupant gesture perception, wheelchair state perception and environment perception experiments, and then completed wheelchair control experiments, including gesture movement control experiments and remote control experiments; perception data can also be viewed on the app.

### 5.2. Gesture Recognition and Wheelchair Control

Figure 18 shows the installation position of the gesture recognition sensor. During the experiment, the wheelchair user sits in the wheelchair and makes five random gestures during the movement: forward, backward, left, right and waving. The embedded processor drives the gesture recognition sensor to detect the gesture change signal and controls the motor drive module to control the two movement wheels of the wheelchair to achieve forward movement, backward movement, left movement, right movement and stopping of the wheelchair.

The wheelchair is controlled by differential speed when turning and the speed of the two moving wheels is always fixed to ensure the safety of the wheelchair turning. In addition, the wheelchair stops moving when the experimenter makes a waving gesture. The experimental process of the four gestures and wheelchair movement is shown in Figure 19.

### 5.3. Wheelchair Posture Awareness

The wheelchair is equipped with the MPU6050 attitude sensor made by Alientek, Guangzhou, China, which can output real-time attitude angle data during the movement of the wheelchair driven by the embedded processor, specifically including pitch, roll and yaw angles. The three-dimensional coordinates output by the sensor can be used to generate a real-time schematic of the wheelchair’s three-dimensional angle changes on a computer, as shown in Figure 20, which shows the three-dimensional coordinates of the wheelchair in a horizontal position and a physical drawing. Figure 21 also shows the three-dimensional coordinates of the wheelchair with the front wheels lifted and a physical drawing. As can be seen, the system illustrates the change in posture angle in a graphical form and the user can clearly understand the real-time safety status of the wheelchair.

### 5.4. Road Environment Sensing

A 3D lidar is installed on the wheelchair and relies on the Robot Operating System to capture the laser point cloud in front of the wheelchair as it moves. During the capture process, the road environment sensing algorithm filters, classifies and identifies the laser point cloud and outputs the results for different objects such as people, cars and trees.

Figure 22 shows the results of the road surface awareness experiment. The top left image is the physical image we took during the experiment, the bottom left image shows the point cloud collected by the lidar and the vehicle recognition results, marked with boxes; the right image shows the recognition results of lane lines, vehicles and other obstacles. By recognising the lane lines, the wheelchair control system can guarantee that the wheelchair can travel on the safe lane. In addition, experiments have shown that the target classification accuracy and target recognition accuracy within 50 m reached over 85% and 95%.

### 5.5. Remote Control Experiment

We installed the NXP RT1062 embedded development board, ESP8266 WiFi module, sensor module and motor driver module on the wheelchair so that the wheelchair side can access the Tencent IoT Explorer as an MQTT client to subscribe and publish data. Figure 23 shows a schematic diagram of the modules on the wheelchair side. Using the various modules shown in the diagram, it is possible to receive remote control commands and publish real-time sensory data.

Figure 24 shows the mobile APP interface developed based on Tencent Connect, including: (1) control buttons for backrest lift and footrest lift; (2) data viewing bar, showing the wheelchair start-up timer, longitude, latitude, speed, lighting brightness, temperature, humidity, and attitude angle information such as pitch, roll and yaw angle. The mobile app also acts as an MQTT client, issuing control commands to the Tencent IoT Explorer, which acts as an MQTT server, and subscribing to wheelchair-aware data.

Figure 25 shows the wheelchair footrest and backrest remote lifting and lowering control experiment based on the above mobile phone APP. The backrest is raised and lowered in the same way as the footrest, and when the backrest lowering button is clicked, the backrest of the wheelchair is automatically lowered.

## 6. Conclusions

In order to improve the perception performance and control of traditional wheelchairs so that they can be operated by different types of users while their loved ones can view the status information of the wheelchair remotely, we have studied the multi-mode perception and control technology of intelligent wheelchairs and explored the remote APP control technology and data transmission technology of wheelchairs based on IoT technology with the following main functions.

(1) Wheelchair perception: we designed and implemented occupant perception, state perception and environment perception for the wheelchair and compared it with correct data after the experiment to test the perception capability. The gesture perception recognition speed is 240 Hz, and can quickly recognise 9 gestures of the user; state perception achieves the acquisition of positioning, velocity and attitude data, relying on BeiDou navigation technology, with a positioning accuracy of around 1 m, and a resolution of 16384LSB/g (Max) and 131LSB/(°/s) (Max) for acceleration and gyroscope, respectively, when collecting attitude. Environmental sensing included road surface sensing and meteorological sensing, in which the accuracy of target classification and target recognition within 50 m reached over 85% and 95%, respectively, and the accuracy of humidity and temperature sensing was ±5%RH and ±2 °C, respectively.

(2) Wheelchair control: we have implemented joystick control, gesture control and APP remote control of the wheelchair. The three control methods can be applied by different people and realize the forward, backward, left and right turn functions of the wheelchair. In the APP, the user can also control the footrest and backrest of the wheelchair and view the real-time sensory data of the wheelchair. In addition, the delay for remote control is around 100 ms, and the accuracy rate of hand gestures and APP control is about 90%.

The intelligent wheelchair developed in this paper has improved the sensing capability of the wheelchair, using the sensing data to also enable the safety monitoring of the wheelchair. In addition, the remote control app allows the user’s loved ones to access the status of the wheelchair and provide assistance in case of danger.

The intelligent wheelchair described in this paper has the advantages of higher intelligence, various operation methods and higher safety performance. After our experiments, the following shortcomings remain: (1) a large number of sensors are installed on the wheelchair and the wire layout is confusing and still needs to be integrated; (2) most of the electrical components are not waterproof, so a waterproof device needs to be designed. In the future, we need solve these two problems. In addition, when the wheelchair is going up and down a slope, the wheelchair may fall due to factors such as speed. We can adjust the horizontal posture of the seat according to the posture information in this paper to ensure safe performance and a safe experience of using the wheelchair.

## Figures and Tables

**Figure 1 micromachines-13-01108-f001:**
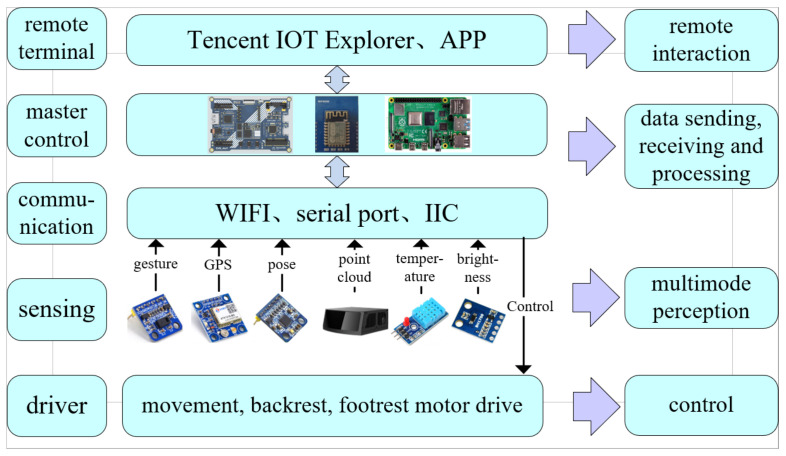
Intelligent wheelchair architecture design.

**Figure 2 micromachines-13-01108-f002:**
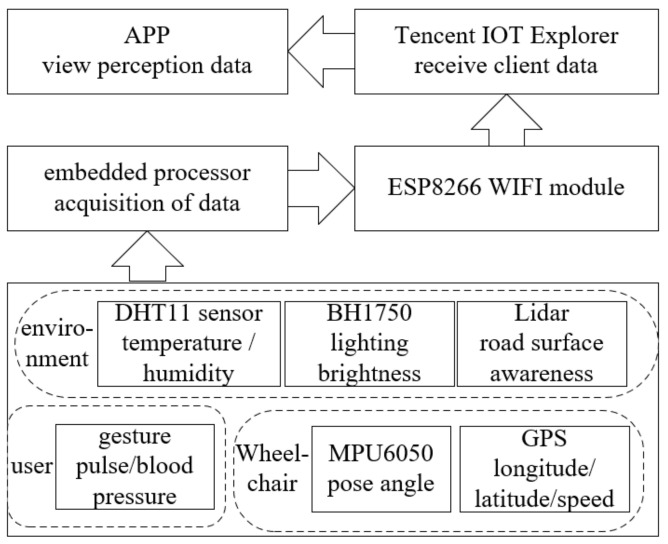
System multimode sensing architecture.

**Figure 3 micromachines-13-01108-f003:**
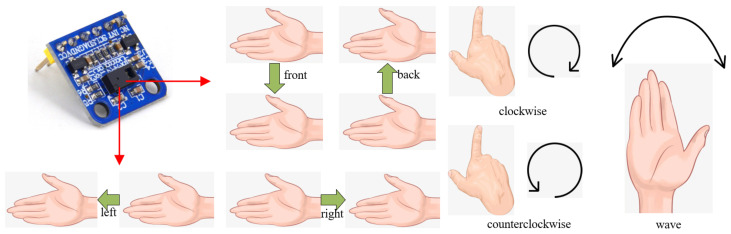
Gesture recognition sensor and gestures.

**Figure 4 micromachines-13-01108-f004:**
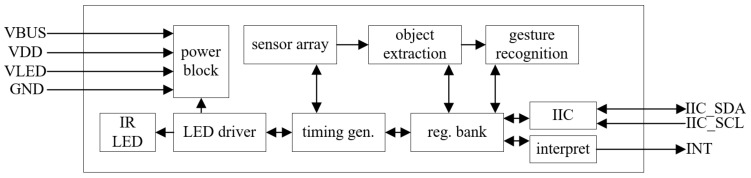
Functional block diagram of the sensor.

**Figure 5 micromachines-13-01108-f005:**
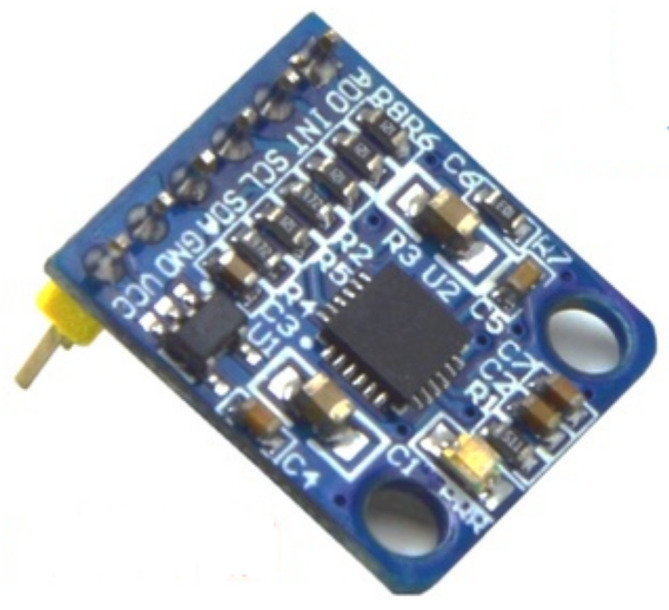
Coordinate diagram of attitude sensor.

**Figure 6 micromachines-13-01108-f006:**
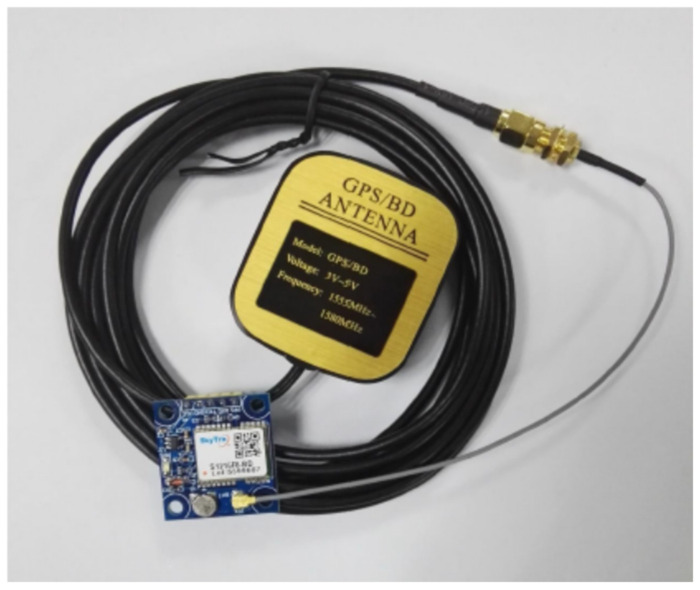
Positioning module.

**Figure 7 micromachines-13-01108-f007:**
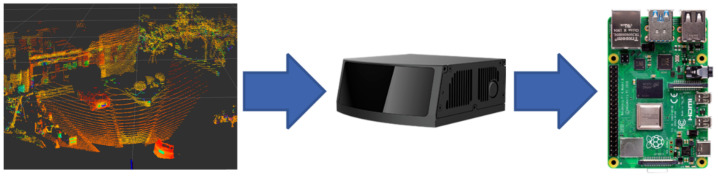
Three-dimensional lidar.

**Figure 8 micromachines-13-01108-f008:**
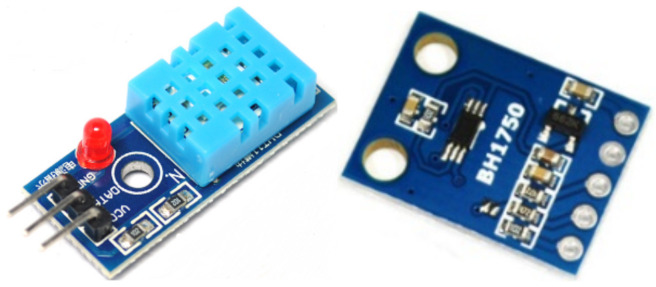
Physical view of DHT11, BH1750 sensor.

**Figure 9 micromachines-13-01108-f009:**
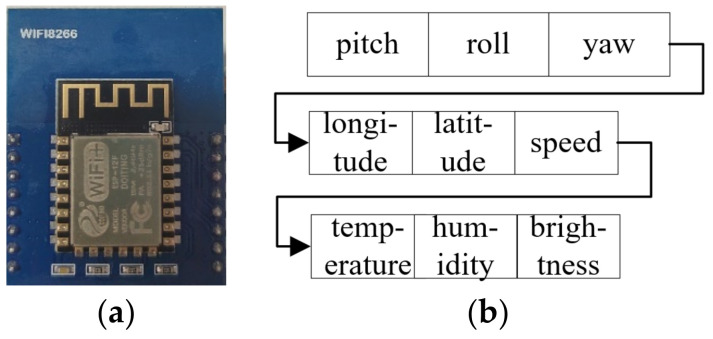
Data package with WiFi module.

**Figure 10 micromachines-13-01108-f010:**
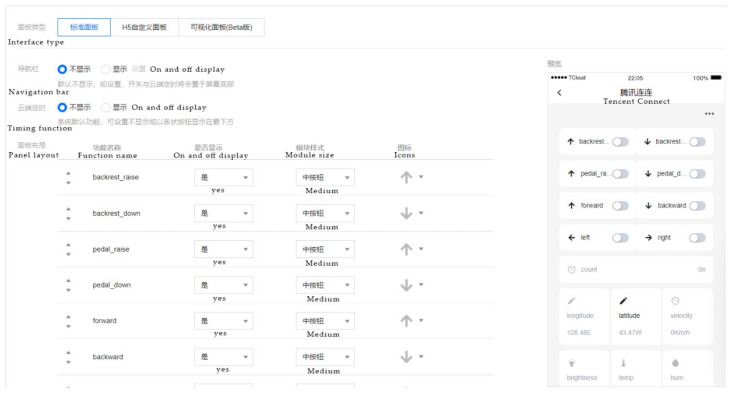
APP interface design.

**Figure 11 micromachines-13-01108-f011:**
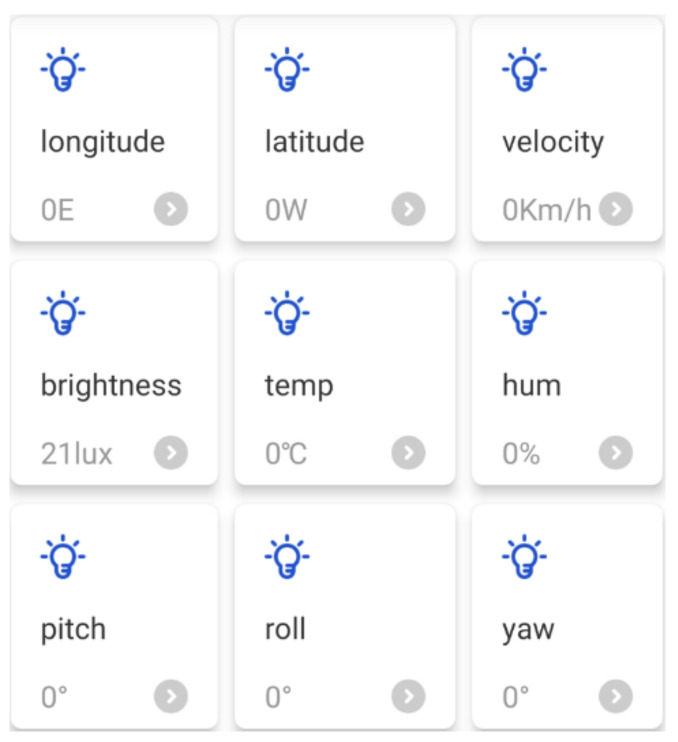
APP interface design.

**Figure 12 micromachines-13-01108-f012:**
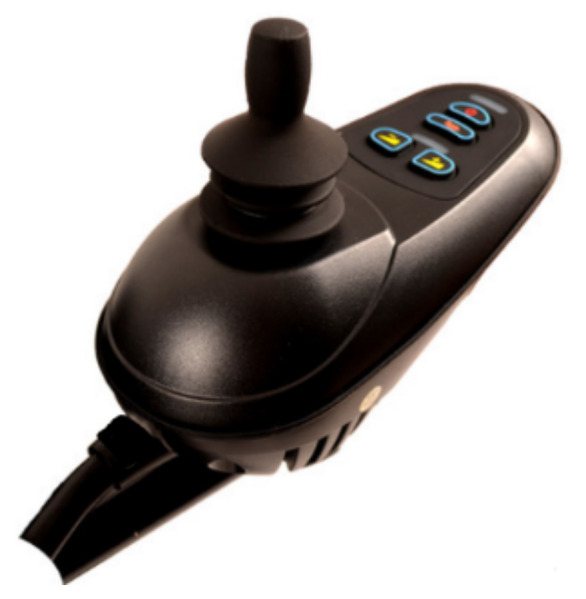
Motorised rocker.

**Figure 13 micromachines-13-01108-f013:**
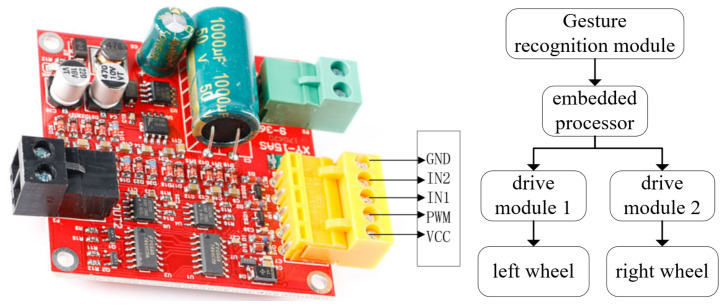
Motor drive module and connection schematic.

**Figure 14 micromachines-13-01108-f014:**
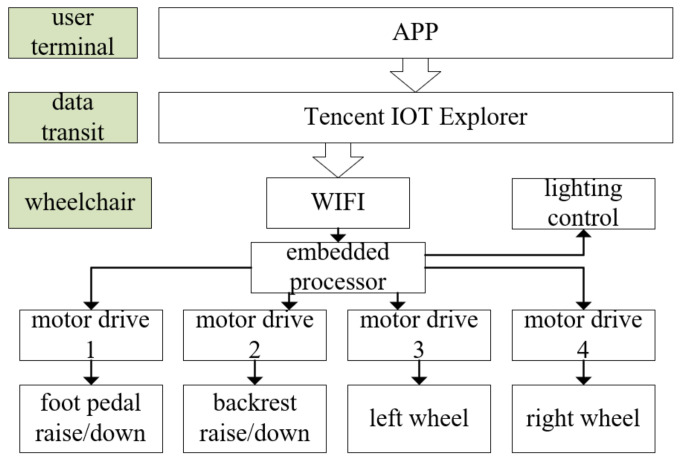
Wheelchair remote control scheme.

**Figure 15 micromachines-13-01108-f015:**
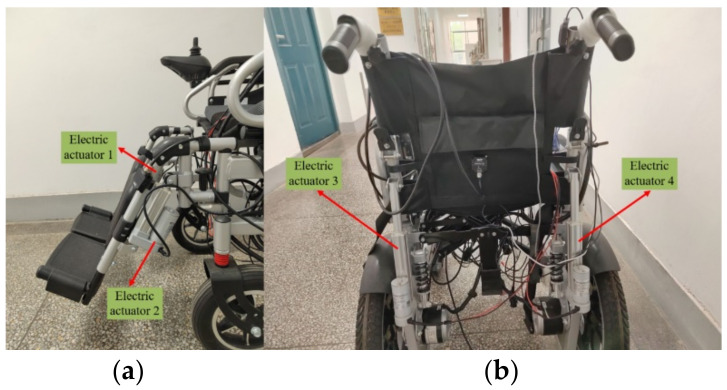
Diagram of the footrest and backrest installation.

**Figure 16 micromachines-13-01108-f016:**
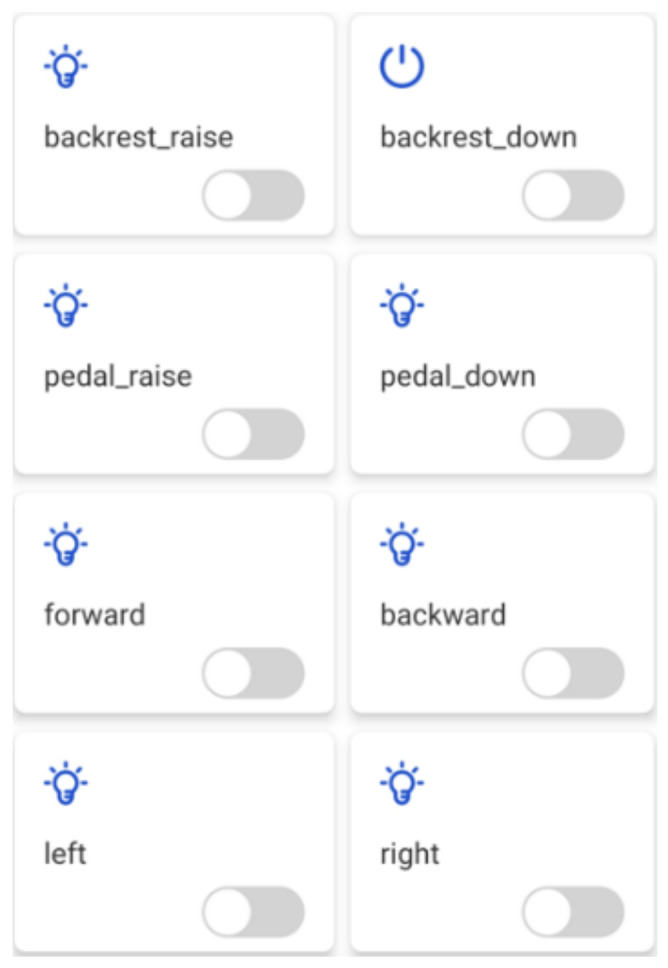
Wheelchair control interface.

**Figure 17 micromachines-13-01108-f017:**
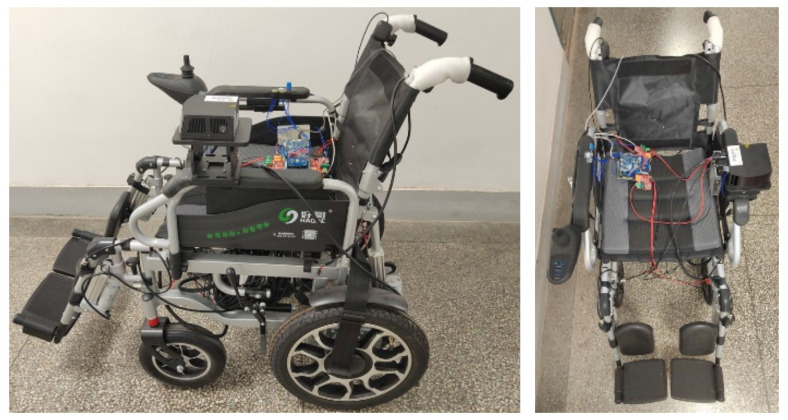
Wheelchair physical appearance.

**Figure 18 micromachines-13-01108-f018:**
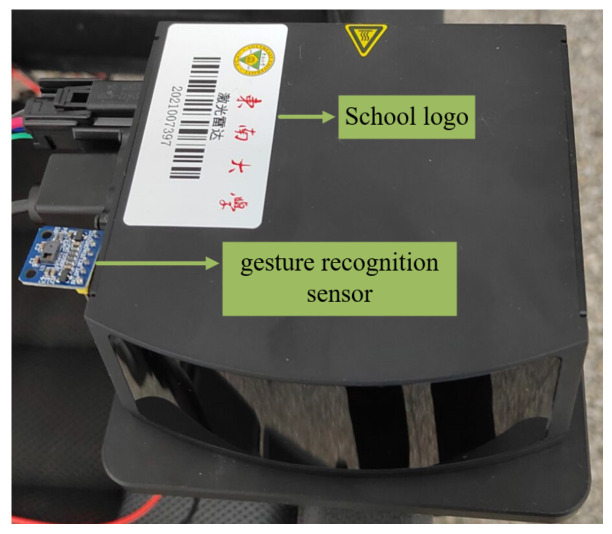
Gesture recognition device.

**Figure 19 micromachines-13-01108-f019:**
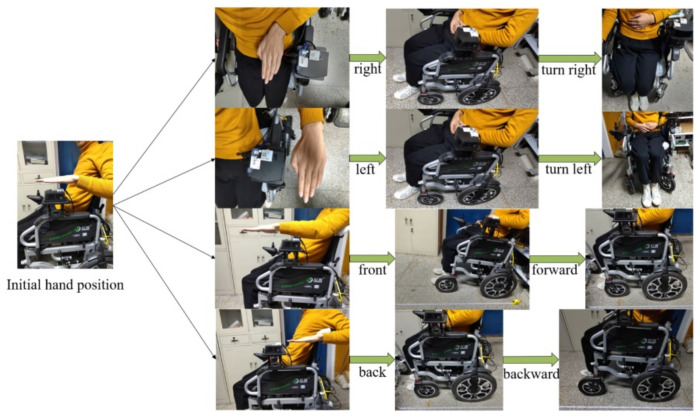
Wheelchair gesture recognition and movement control.

**Figure 20 micromachines-13-01108-f020:**
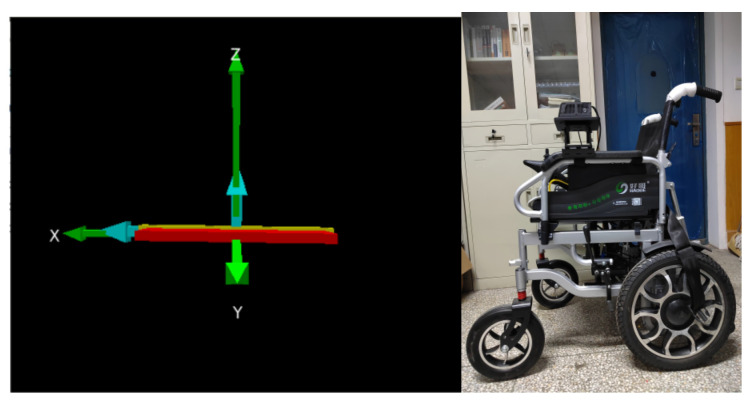
Wheelchair in horizontal position.

**Figure 21 micromachines-13-01108-f021:**
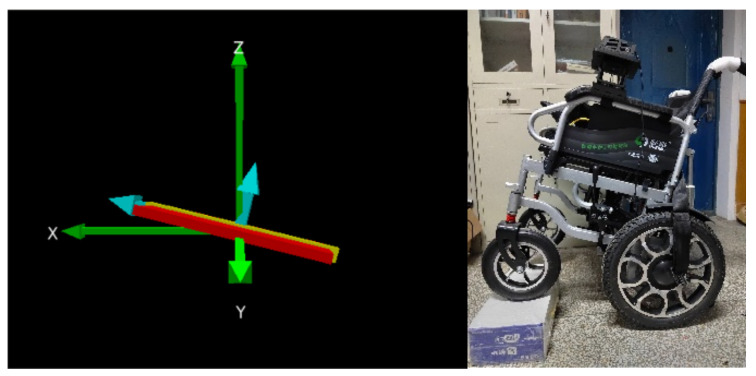
Wheelchair tilt condition.

**Figure 22 micromachines-13-01108-f022:**
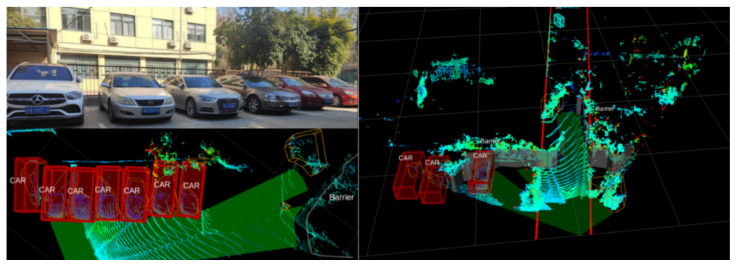
Road object perception experiment.

**Figure 23 micromachines-13-01108-f023:**
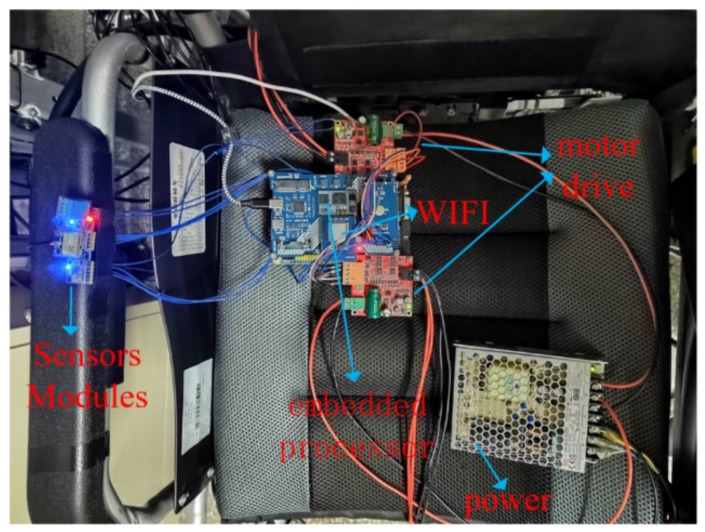
Remote control module.

**Figure 24 micromachines-13-01108-f024:**
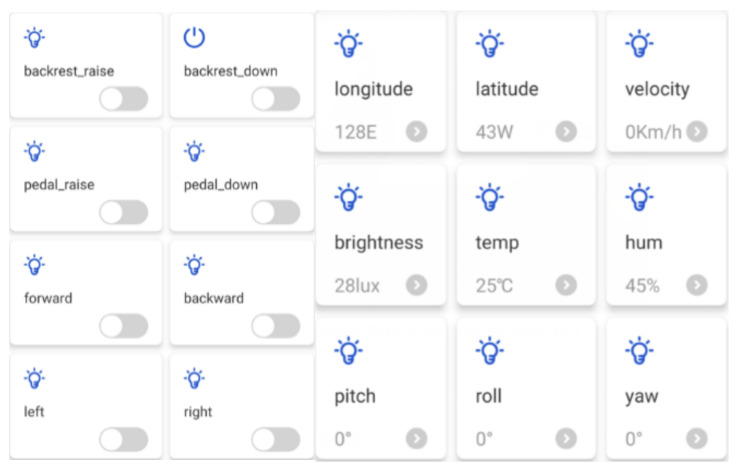
Mobile phone APP interface.

**Figure 25 micromachines-13-01108-f025:**
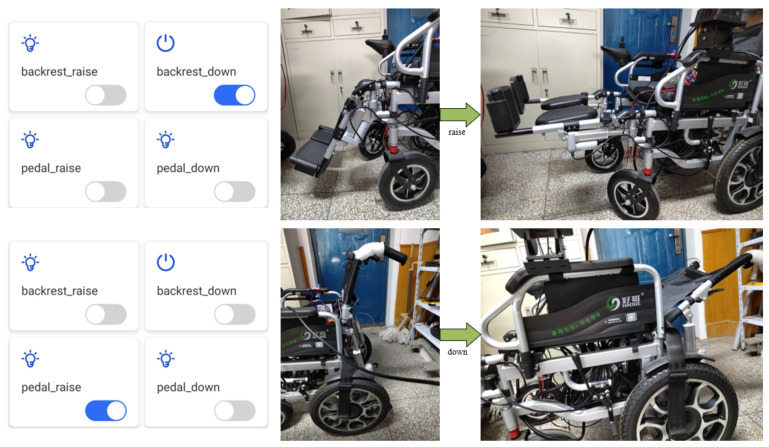
Wheelchair remote control experiment.

**Table 1 micromachines-13-01108-t001:** Lidar parameter table.

Radar Performance	RS-LiDAR-M1
Ranging capability	0.5–200 m (150 m, 10% NTST)
Accuracy	±3 cm, (0m~100m)
Viewing angle (vertical)	25° (−12.5°~+12.5°)
Viewing angle (horizontal)	120° (−60.0°~+60.0°)
Angular resolution	0.2°
Frame rate	10 Hz

**Table 2 micromachines-13-01108-t002:** Communication topics.

Topic	Content
ID/${deviceName}/data	Sensory data
ID/${deviceName}/event	Event reporting
ID/${deviceName}/event	Control commands
Viewing angle (horizontal)	120° (−60.0°~+60.0°)
Angular resolution	0.2°
Frame rate	10 Hz

## Data Availability

The data presented in this study are available on request from the corresponding author.

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
