# Peer review of "IoT Wheelchair Control System Based on Multi-Mode Sensing and Human-Machine Interaction"

_micromachines, 2022, doi:10.3390/mi13071108_

Round 1

Reviewer 1 Report

1. General overview

The article presents the results of work on the design, construction and testing of an intelligent wheelchair. The Authors equipped the wheelchair with a microcontroller, number of sensors and developed a control program with an application that allows one to view data and control the device. with the means of a remote device. The aim was to develop a solution that could be used by many different users and their guardians, regardless of the degree of motor disability.

The article is divided into many sections, each of which describes a different aspect of the functionality of the presented device. The structure of the text seems logical and understandable. Some errors related to the translation and editing of the text can be identified. In general, I rate the article presented to me for review positively and I believe that it is suitable for publication after minor corrections.

2. Detailed comments

- Line 77: “Fahd N. Al-Wesabi et al combined…” - the seems to be a repetition in this line;

- Line 141-143 - whole point (2): please consider rephrasing this sentence as it is not quite understandable;

- Lines: 246, 347, 377: The Authors use the form of “the article designs/this article…” which is grammatically incorrect and, I think, results from translation errors. Please consider rephrasing in the form of: “In this paper Authors designed/As a part of presented work [...] was designed”or similar;

- Figure 10: Fig 10 is completely unreadable. It seems to me that it should be bigger and of better quality;

I propose adding fragments to the manuscript in order to answer the following questions:   

- Was a commercially available wheelchair used, or was the frame and other mechanical components designed specifically for the purpose of building a prototype?

- With what frequency the measurements from the sensors take place during the normal operation of the device (in some places it is indicated that the frame rate is 10 Hzin the case of some sensors, but nothing else)?

- It seems to me that a large number of sensors require a certain computing power. In addition, wireless communication, positioning, display, etc. What is the electrical power demand of these solutions, eg maximum and during typical use?

-  What follows from the above: the battery of what capacity has been installed in the device, for which the expected operating time will be sufficient between charges? 

- How much did the weight of the device increase after installing all the accessories (electronics + battery, etc)?

Reviewer 2 Report

This work presents an intelligent wheelchair based on multimodal sensing and human-machine interaction. A multimodal sensing scheme is designed (including occupant gesture sensing, wheelchair state sensing, and environment sensing), then human-machine interaction-based wheelchair control methods are investigated, and wheelchair remote control technology based on mobile phone APP, Tencent IOT Explorer and WiFi module is explored.  Experimental validation was performed on the prototype.

The provided information is relevant for the knowledge field. Nevertheless, some issues should be addressed before this manuscript could be considered for publication.

 1) The structure of the manuscript should be improved, including Materials and Methods and Results sections.

2) The Results should clearly present the evaluation methodology and the quantitative results.

3) The scientific contribution of this work is not clear, should be highlighted from the state of the art.

4) The Conclusion section should include quantitative results, advantages and disadvantages, limitations, and recommendation for real implementations.

5) Future work recommendations should be provided.

The structure of the manuscript, methodology and results are not acceptable. The manuscript seems to be a technical report, it should be rewritten focused on scientific aspects, providing more information on theoretical fundamentals, and experimental validation. Furthermore, the scientific contribution of this work is not clear

Reviewer 3 Report

1.            It is recommended to improve the quality of Figures 3, 7, 10.

2.            It is recommended to explain in the text all the symbols and parameters that are used in formula (1).

3.            It is recommended to indicate from which literary source the data in Table 1 were taken.

4.            It is recommended to describe in more detail what is shown in Figure 22.

5.            It is recommended to decrypt all abbreviations used, for example: MIT, APP, LIDER, AEG, LED, MEMS, DC.

6.            It is recommended to specify the manufacturer and provide a link to the description of all sensors used, for example: DHT11, BH1750.

7.            It is recommended to describe the test results in more detail. How many people participated in the testing? What is the received reliability of the wheelchair control system under various conditions?

8.            How are you planning to implement the developed wheelchair control system? How are you planning to continue this research?

Round 2

Reviewer 2 Report

The authors addressed the recommendations, the manuscript has been sufficiently improved and could be considered for publication.